# Eye-Tracker Analysis of the Contrast Sensitivity of Anomalous and Normal Trichromats: A Loglinear Examination with Landolt-C Figures

László Sipos [1], Attila Gere [1], Zoltán Kókai [1], Ákos Nyitrai [1], Sándor Kovács [2,*], Ágnes Urbin [3], Krisztián Samu [3] and Klára Wenzel [3]

1   Department of Postharvest, Supply Chain, Commerce and Sensory Science, Institute of Food Science and Technology, Hungarian University of Agriculture and Life Sciences, 39-43 Villányi út., 1118 Budapest, Hungary; Sipos.Laszlo@uni-mate.hu (L.S.); Gere.Attila@uni-mate.hu (A.G.); Kokai.Zoltan@uni-mate.hu (Z.K.); Nyitrai.Akos.Gabor@hallgato.uni-szie.hu (Á.N.)
2   Department of Economical and Financial Mathematics, Institute of Statistics and Research Methodology, Faculty of Economics and Business, University of Debrecen, 138. Böszörményi út., 4032 Debrecen, Hungary
3   Department of Mechatronics, Optics and Mechanical Engineering Informatics, Faculty of Mechanical Engineering, Budapest University of Technology and Economics, 3 Műegyetem rkp., 1111 Budapest, Hungary; urbin@mogi.bme.hu (Á.U.); samuk@mogi.bme.hu (K.S.); wenzel@mogi.bme.hu (K.W.)
*   Correspondence: kovacs.sandor@econ.unideb.hu; Tel.: +36-204-235-361





**Featured Application: Selected and expert sensory assessors should not have any deficiencies that could affect their perception and adversely affect their sensory performance and thus affect the reliability of their evaluations. In sensory practice, numerous pseudoisochromatic images have been developed to test color vision, and the anomaloscope is used instrumentally. The eye-tracking and series of achromatic and colored pseudoisochromatic tests used in our research, and their combination, provide detailed analytical possibilities for the eye movements of the subjects. By testing the achromatic and colored figures, the decision and decision time of groups with different contrast sensitivity can be evaluated together with the survival analysis. The combined method can also be used for diagnostic testing of color vision.**

**Abstract:** The contrast sensitivity of normal and anomalous trichromats were examined with Landolt-C figures by eye-tracking system. For the measurements, two series of test images (achromatic and colored) were designed. The difficulty levels of the tests were gradually increased after each right answer. In the case of the observation of the ring of the Landolt-C figures, the variables related to fixation duration, fixation count, visit duration and count significantly affected this subject, success or image parameters, and their interactions. The main questions of this study were as follows: Which statistical method is suitable to model the differences between anomalous and normal trichromats? Which eye-movement variables have a significant effect on the investigated parameters and on their interactions? Is there any significant difference between eye-movement variables of normal and anomalous trichromats? How does the survival time of anomalous and normal trichromats change in the case of achromatic and colored figures? The results showed that the right answers of anomalous and normal trichromats can be described with multiple or cross-classified contingency tables evaluated effectively by loglinear regression. The survival analysis showed that normal trichromats are more successful in interpreting colored images, while anomalous trichromats seemed to be more efficient in perceiving achromatic images.

**Keywords:** color-vision deficiency; Landolt-C; eye-tracker; loglinear regression; Scheirer–Ray–Hare test

## 1. Introduction

Eyes play an important role in our life, not only in seeing objects of the environment, but also in reading books, watching films, or recognizing and discriminating objects and members [1]. The visual acuity of the eye is generally regarded as the most important factor for seeing objects. Objects can generally be better distinguished from each other or from their background if the luminance or color difference is large. Contrast describes the difference in appearance of two or more parts of a field seen simultaneously or successively, a metrics for luminance or color differences. Contrast sensitivity is the capacity of the eye to perceive this difference [2,3]. Vision is created by the contrast sensitivity of the eye and the image processing of the brain [4–6].

The effect of color vision on food and commercial product judgements is an important topic of international research [7–15]. As an example, the effect of background color and pattern that can be traced back to as contrast between chromaticities of the two defined visual area is observed in detail [16–18]. Another aspect of color vision in food industry is the effect of colors on sensory evaluation [15]. Requirements regarding the visual acuity and color vision of sensory assessors are detailed in standards [19,20]. In the case of visual evaluation, preliminary testing for anomalous color vision (color-vision deficiency) is required. Even though there are different types of color-vision deficiencies, a common phenomenon is the reduced ability in chromatic discrimination and contrast sensitivity, compared to normal trichromats [21].

Even understanding individual differences between normal trichromat observers can be important to provide a standardized evaluation process [22,23]. With the involvement of anomalous trichromats, more significant difference may occur, justifying the need for further research.

In international researches the most accepted and accurate method for identification of color-vision deficiency is the anomaloscope examination. However, there are many pseudoisochromatic methods applied in practice as well [24–29]. During these examinations, observers have to find differences or parity in special patterns the temporal effect of chromatic adaptation should be considered also.

Observation of observers' gazing behavior with eye tracking is a widely used method that can provide detailed analysis based on the wide range of measured parameters, such as the count or the duration of fixations or the decision time [30–32].

Therefore, the aim of this research was to answer whether eye tracking can be a method for testing color-vision deficiency.

The main question of this research was whether anomalous and normal trichromats can be distinguished based on eye movements under different conditions of luminance and chromatic contrast sensitivity. In line with this question measurement with pseudoisochromatic figures were set in order to detect temporal patterns related to the results of the tests. Our main research questions were as follows:

1.　Which statistical method is suitable to model the differences between the number of right answers of anomalous and normal trichromats?
2.　Which of eye movement variables have significant effect on the parameters (image, success and subject) and their interactions?
3.　Is there any significant difference between eye movement variables of normal and anomalous trichromats?
4.　How does the survival time of anomalous and normal trichromats change in the case of achromatic and colored figures?

## 2. Materials and Methods

### 2.1. Experiment Structure

#### 2.1.1. Visual Stimuli

For the eye-tracker measurements two series of test images were designed. One of them contains achromatic (dark pattern on light background), while the other contains

colored (red pattern on green background) pseudoisochromatic images. In both series, the patterns trace out Landolt-C figures, and the task was to find the orientation of them [33,34].

In the images of the achromatic series, the Landolt-C figure is dark gray, while the background is light gray. The size and brightness of the dots are slightly different in a random layout like in the figures of the Ishihara test. The luminance contrast progressively decreases on the images of the series and the Landolt-C figures are getting lighter. The series contains 10 images. The first image of the series is shown on the left part of Figure 1. The luminance contrast values between the Landolt-C figures and the backgrounds can be found in Table 1.

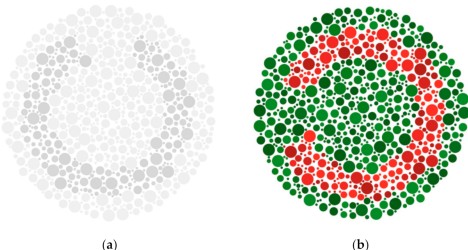

Figure 1. The first images of the test series: (**a**) achromatic and (**b**) colored.

**Table 1.** The contrast values of the applied pseudoisochromatic images.

| Colored Images | | Achromatic Images | |
|---|---|---|---|
| Ordinal Number | Chromatic Contrast, $C_{achr}$ % | Ordinal Number | Luminance Contrast, $C_{chrom}$ % |
| 00 | 1.98 | 16 | 0.00 |
| 01 | 2.71 | 17 | 1.03 |
| 02 | 2.50 | 18 | 2.07 |
| 03 | 5.19 | 19 | 3.10 |
| 04 | 7.03 | 20 | 4.14 |
| 05 | 10.54 | 21 | 5.18 |
| 06 | 13.20 | 22 | 6.22 |
| 07 | 15.20 | 23 | 7.27 |
| 08 | 16.22 | 24 | 8.31 |
| 09 | 20.25 | 25 | 8.84 |
| 10 | 21.62 | | |
| 11 | 25.67 | | |
| 12 | 28.98 | | |
| 13 | 32.00 | | |
| 14 | 34.89 | | |
| 15 | 37.78 | | |

The colored series contain pictures with pseudoisochromatic figures in which the dots of the background are green, while the Landolt-C figures are built up from red dots. The size and brightness of the dots are slightly different in a random layout. The colored series contains 16 images following the red–green confusion lines. The chromatic contrast between the Landolt-C figure and the background is the higher in the first image and decreases solidly as more and more green is mixed to the color of the Landolt-C. The chromatic contrast values between the Landolt-C figures and the backgrounds is found in Table 1. The pictures were organized in two groups consisting of 16 colored and 10 grayscale pictures.

Both the achromatic and the chromatic contrast between the pattern and the background were large in the first images and low in the last ones; hence, the images were ordered from the easiest to the most difficult one. The last ones were designed, as the perceived differences between the background and the Landolt-C figure were smaller than the just noticeable stimuli of normal trichromats in the state of adaptation defined by the

chromaticity of the background; even normal trichromats could not find the orientation of the Landolt-C. The ordinal number of the first image where the subject could not tell the orientation of the Landolt-C was the variable that represented the result of the measurements [35]. This number provides representative information about the actual subject's contrast sensitivity [3].

The calibration of the eye-tracker display with which we determined the chromatic and luminance contrast of the images was executed, using an X-rite Eye-One pro device. The display of the eye-tracker measurement was calibrated for sRGB (gamma = 2.2, Correlated Colour Temperature (CCT) = 6500 K) Cathode-ray tube (CRT) monitor (Samsung SyncMaster 757 MB, SN: PU17HSAX907276V). The stimulus images were shown on the eye-tracking display (17 in., 1280 × 1024 pixel resolution). X, Y and Z tristimulus values were determined with sRGB to XYZ conversion based on the R, G and B values of the images. Further on L, M and S values were calculated, that are necessary to determine the chromatic and luminance contrast. The following transformation was applied, based on the average X, Y and Z tristimulus values of the background and the Landolt-C figures [36]:

$$
\begin{bmatrix} L \\ M \\ S \end{bmatrix} = \begin{bmatrix} 0.15514 & 0.54321 & -0.03286 \\ -0.15514 & 0.45684 & 0.03286 \\ 0 & 0 & 0.00801 \end{bmatrix} \cdot \begin{bmatrix} X \\ Y \\ Z \end{bmatrix}
$$

Two pictures were used as warm-up to familiarize the subjects with the procedure that were not included in the data analysis. This took an average of 4–6 s. The duration of responses to pseudo-isochromatic images was participant-dependent. On average, 25–35 s was spent in total. There was no break between the familiarization, the chromatic and the achromatic sessions. For each task, the images were arranged in the middle of the screen, and a fixation cross between the tasks was displayed to standardize the zero point between the presentations of stimuli. Two main areas of interest (AOIs) were defined: the ring and the hiatus of the Landolt-C figure. The presentation of colored and achromatic series was rotated between subjects, although the picture order within the series was always the same.

### 2.1.2. Subjects

Ninety subjects (42 males and 48 females) aged between 18 and 35 (25.5 ± 3.6 years) participated in the study. Normal and defective color vision was confirmed with anomaloscopic measurements: R/G scores within the range of 40 ± 5 indicated normal color vision, while R/G scores excluding this range indicated defective color vision. Eighty-one of the subjects were normal trichromats and nine of the subjects reported color vision deficiency which was confirmed with the anomaloscopic measurements. Since in many cases subjects with deuteranomaly have similar results on the applied pseudoisochromatic tests than of normal trichromats, we excluded deuteranomalous subjects and we invited only subjects with protanomaly. Visual acuity of the subjects was monitored in a report and tested with the first test figures of both test series. Since the field of view under which the detectable stimulus was shown did not change during the test, we used the first images of the two test series. In case subjects could clearly identify the orientation of both Landolt-C characters, we accepted that their visual acuity is sufficient for participation.

The measurements were recruited at Szent István University, Budapest, Hungary. Within the first contact with subjects the method, the necessary time capacity and the test procedure were explained in detail. The study was performed in accordance with the ethical guidelines for scientific research of Szent István University. Subjects were informed that they could withdraw themselves and their data from the study without giving an explanation, at any time. All subjects agreed to these conditions and participated without receiving a reward for their participation.

### 2.1.3. Eye-Tracking Procedure

A Tobii X2-60 eye-tracker and Tobii Studio software (version 3.0.5, produced by Tobii Technology AB, Stockholm, Sweden) were used for recording and analyzing the gazing

behavior of the 90 subjects. The experiment took place under controlled environment (lighting, temperature etc.) in the Sensory Laboratory of the Department of Postharvest and Sensory Evaluations of Szent István University, Budapest, Hungary.

Lighting conditions followed the recommendations of the ISO 3664 and ISO 12646 Standards [37,38]. Approximately homogeneous luminance of 300lx was provided with a special attention on excluding glare and disturbing visual elements in the field of view.

Subjects took a seat in front of the eye-tracker screen in a relaxed way with their eyes at a distance of approximately 70 cm holding the mouse in their dominant hand. They were instructed not to change their position during the test. Measurements were executed binocularly. After successful calibration, an instruction screen explained the details of the measurement. A black fixation cross was shown for 3 s in the middle of the screen to standardize the zero point of the subjects. Without time limit, subjects had to find the gap in the Landolt-C figure. Subjects had to click the left mouse button once as soon as they found the gap, then the mouse pointer appeared on the screen and they had to click on the gap. Two warm up tasks were used to familiarize subjects with the procedure and these were left out from the data analysis.

### 2.2. Applied Statistical Methods and Analysis of Success

Moving from symmetrical to asymmetrical, we applied loglinear analysis. This analysis models the log odds that a given observation (fail or success) will appear in one category (normal trichromats) of a dependent variable relative to other categories (anomalous trichromats). The purpose of the analysis is to model the asymmetry that exists in the data and translate it in terms of odds.

Loglinear analysis requires fewer distributional assumptions and limitations and can be applied in any circumstance in which we study the relationships between categorical variables. In loglinear models, the cell probabilities for the cross-classified I × J × K contingency table are decomposed into multiplicative effects for each variable and for the associations among them [39] and subsequently the logarithms of the cell frequencies are formulated as a linear function of the estimated parameters. In this context, loglinear analysis is an equivalent to analysis of variance (ANOVA) with categorical independent variables and the dependent variable is of the logarithm of the cell probabilities.

As the algorithm takes the natural logarithm of these cell probabilities, large number of observation is required so as to avoid zero frequencies and observations should be obtained in the same circumstances independently from each other. In our study, a general nonhierarchical log-linear model with three categorical variables and a continuous covariate was fitted. Data were analyzed, using SPSS 23 software, using LOGLINEAR module, and are given in the following table (Table 2).

**Table 2.** Contingency table of subjects and success.

| | | | | Subject (A) | |
| --- | --- | --- | --- | --- | --- |
| | | | | **Normal Trichromat** | **Anomalous Trichromat** |
| Success (C) | Failed | Image (B) | Achromatic | 72 (93.5%) | 5 (6.5%) |
| | | | | 173 (79.4%) | 45 (20.6%) |
| | Succeeded | Image (B) | Colored | 252 (88.4%) | 33 (11.6%) |
| | | | | 460 (92.4%) | 38 (7.6%) |

In this 2 by 2 by 2 contingency table, 1078 observations are cross-classified on three variables: subject (A) with levels $i$ ($i = 1$ for normal trichromats; 2 for anomalous trichromats), image type (B) with levels $j$ ($j = 1$ for achromatic; 2 for colored) and success (C) with levels $k$ ($k = 1$ for failed; 2 = succeeded). Suppose further that we have a continuous covariate which is the ordinal number of the cards (D) (its value varies from 0 to 25). An important feature of the LOGLINEAR module is the possibility of involving continuous

covariates to the model. The more parsimonious model obtained can be denoted as follows (Equation (1)):

$$\ln\left(m_{ijk}\right) = \lambda_0 + \lambda_{ijk}^{ABC} + \lambda_{jk}^{BC} + \lambda_k^C \times \theta_{ijk}^D \tag{1}$$

where $m_{ijk}$ is the expected frequency for cell $(i,j,k)$ and each of the three subscripted $\lambda$-terms sums to zero over each lettered subscript. For instance, we have the following (Equation (2)):

$$\sum_i \lambda_{ijk}^{ABC} = \sum_j \lambda_{ijk}^{ABC} = \sum_k \lambda_{ijk}^{ABC}; \lambda_{11}^{BC} = -\lambda_{12}^{BC} = -\lambda_{21}^{BC} = \lambda_{22}^{BC} ; \lambda_1^C = -\lambda_2^C \tag{2}$$

and so on. For each pattern of the classifying variables the mean value of covariate D is calculated (denoted by $\theta_{ijk}^D$) and built into the design matrix for model parameter estimation [40]. The grand mean term ($\lambda_0$) is a normalising constant estimated to make cell probabilities sum to 1, according to the following formula [39] (Equation (3)):

$$\lambda_0 = -ln \sum_{ijk} \exp\left(\lambda_{ijk}^{ABC} + \lambda_{jk}^{BC} + \lambda_k^C \times \theta_{ijk}^D\right) \tag{3}$$

The $\lambda$-terms can easily be converted to odds ratio as follows [41] (Equation (4)):

$$\lambda_{111}^{ABC} = \frac{1}{8} \times \ln\left(\frac{m_{111} \times m_{221}}{m_{121} \times m_{211}} \times \frac{m_{122} \times m_{212}}{m_{112} \times m_{222}}\right) \text{ and } \lambda_{11}^{BC} = \frac{1}{8} \times \ln\left(\frac{m_{121} \times m_{211}}{m_{111} \times m_{221}} \times \frac{m_{122} \times m_{212}}{m_{112} \times m_{222}}\right) \tag{4}$$

In matrix terms, model (1) can be described by the following matrix equation (Equation (5)):

$$ln\left(\overrightarrow{m}\right) = X \times \overrightarrow{b} \tag{5}$$

where $\overrightarrow{b} = (\beta_0, \beta_1, \beta_2, \beta_3)$ is a $4 \times 1$ parameter vector for the terms in model (1), $\overrightarrow{m}$ is a $8 \times 1$ vector of the cell probabilities and X is the $8 \times 4$ design matrix. The first element in the parameter vector is identical to the grand mean; the 4th column of the design matrix includes the covariate means. Model (1) can be written in the following form using formula (Equation (6)):

$$\ln\begin{pmatrix} m_{111} \\ m_{112} \\ m_{121} \\ m_{122} \\ m_{211} \\ m_{212} \\ m_{221} \\ m_{222} \end{pmatrix} = \begin{bmatrix} 1 & 1 & 1 & 23.43 \\ 1 & -1 & -1 & -20.3 \\ 1 & -1 & -1 & 13.37 \\ 1 & 1 & 1 & -8.65 \\ 1 & -1 & 1 & 24.6 \\ 1 & 1 & -1 & -20.76 \\ 1 & 1 & -1 & 9.8 \\ 1 & -1 & 1 & -5.76 \end{bmatrix} \cdot \begin{bmatrix} \beta_0 \\ \beta_1 \\ \beta_2 \\ \beta_3 \end{bmatrix} \tag{6}$$

Scheirer–Ray–Hare (SRH) test is an equivalent of parametric two-way ANOVA test including the interaction terms as well. It is based on ranks and also suitable for any situations where ordinal data are used. On the other hand, the test makes no assumption about the normal distribution of the data. The procedure of the SRH test is as follows [42]:

1. Carry out a standard parametric ANOVA on the ranked data
2. Calculate the mean square (MS) by dividing the total sum of squares by the total degree of freedom
3. Take the relevant sum of squares (SS) of the factors and divide by the value of MS and calculate the SS/MS value for each factor
4. Take the relevant SS/MS value and the degree of freedom (df) of the relevant factor and compute the $\chi^2$ distribution with the parameters of SS/MS and degrees of freedom in order to gain the modified *p*-values

We used SRH test to determine the significant differences in the eye-tracker variables using the significant interaction terms of the loglinear model. These variables were also tested in the loglinear model but did not have any direct influence to the cell frequencies.

The significant combinations of variables (subject × success × image and image × success) were also analyzed in detail by Mann–Whitney test for the parameters of the Eye-Tracker.

Survival analysis was developed for cases where the task is the analysis of the elapsed time until an event occurs. This elapsed time is usually called as survival time [43,44]. In this research the analysis of the survival time requires special methods since the decision times of anomalous and normal trichromats cannot be directly compared because of the lack of normal distribution. The aim of the method is to estimate the survival function and the execution of related significance analyses. The survival function gives that how does the rate of the survivals decrease approximately from the starting point in the function of time [45]. The comparison of the survival diagrams was executed with the generalized Wilcoxon-test [46]. For the survival analysis, Statistica 12.0 software-package (Statsoft Inc. Tulsa, OH, USA) was used.

## 3. Results

### 3.1. Analysis of Success

Observing the parameter estimate outputs, we can see that the parameters that should be integrated in the model can be unequivocally detected. Based on the results it can be stated that the following parameters are significant ($\alpha < 0.05$): subject × success × image, success × image, subject, success and image (see Table 3). Therefore, we excluded the following parameters from the model: subject × success and subject × image.

**Table 3.** Parameter Estimates and their confidence intervals (CI).

| Effect | Parameter | Estimate | SE | Z Score | *p*-Value | Lower CI (95%) | Upper CI (95%) |
|---|---|---|---|---|---|---|---|
| Subject × success × image | 1 | 0.213 | 0.067 | 3.187 | 0.001 | 0.082 | 0.344 |
| Subject × success | 1 | −0.073 | 0.067 | −1.094 | 0.274 | −0.204 | 0.058 |
| Subject × image | 1 | 0.097 | 0.067 | 1.458 | 0.145 | −0.034 | 0.228 |
| Success × image | 1 | −0.281 | 0.067 | −4.204 | <0.001 | −0.412 | −0.150 |
| Subject | 1 | 1.052 | 0.067 | 15.761 | <0.001 | 0.921 | 1.183 |
| Success | 1 | −0.483 | 0.067 | −7.233 | <0.001 | −0.614 | −0.352 |
| Image | 1 | −0.466 | 0.067 | −6.975 | <0.001 | −0.597 | −0.335 |

The loglinear model was built up based on these relevant parameters. Further on the parameter of success was excluded, because the HILOGLINEAR module of SPSS estimates the output of the categorical parameters, but it cannot handle covariant variables. Finally, when the model with the covariant parameters was tested, the effects of success and image were also excluded and only the effect of their interaction remained in the model. Based on the results of the model subject x success and subject x image interactions had no significant effect and were left out from the final model. (The success×ordinal number parameter was included, because as the ordinal number increases, the task is getting harder to complete, hence the rate of success decreases (the coefficient is −0.106; see Table 4).

**Table 4.** Estimates of Parameters and their confidence intervals (CI).

| Parameters | Coefficient | SE | Z Score | *p*-Value | Lower CI (95%) | Upper CI (95%) |
|---|---|---|---|---|---|---|
| Subject | 0.000 | - | - | - | - | - |
| Success | 1.188 | 1.868 | 0.636 | 0.525 | −2.474 | 4.850 |
| Image | 0.000 | - | - | - | - | - |
| Success × ordinal number | −0.106 | 0.114 | −0.929 | 0.353 | −0.330 | 0.118 |
| Success × image | 0.393 | 0.777 | 0.505 | 0.614 | −1.131 | 1.916 |
| Subject × success × image | 0.115 | 0.168 | 0.688 | 0.492 | −0.213 | 0.444 |

Success was not significant, so it was excluded from the final model, which only included the following parameters: success × image, success × ordinal number and subject × success × image. The loglinear model fits the data as well as the full saturated hierarchical model (LR $\chi^2$ = 0.384; *p* = 0.535). In the loglinear model, each three parameter interactions were significant at the 10% level (see Table 5).

**Table 5.** Estimates and confidence intervals of parameters kept in the model.

| Parameters | Coefficient | SE | Z Score | *p*-Value | Lower CI (95%) | Upper CI (95%) |
|---|---|---|---|---|---|---|
| Success × image | −0.102 | 0.054 | −1.876 | 0.061 | −0.208 | 0.005 |
| Success × ordinal number | −0.034 | 0.002 | −14.228 | <0.001 | −0.038 | −0.029 |
| Subject × success × image | 0.218 | 0.054 | 4.019 | <0.001 | 0.112 | 0.324 |

The coefficient of success×image is −0.102. Because of the loglinear model, this value was multiplied by four, and we took the exponential of this value with base *e* (odds ratio = 0.665). We prepared the analysis of the cross-tabulation of the success × image parameter for the validation of that (see Table 6).

**Table 6.** Cross-tabulation of success × image parameters.

| | | | Image | | Total |
|---|---|---|---|---|---|
| | | | Achromatic | Colored | |
| Success | Failed | Count | 77 | 218 | 295 |
| | | % within success | 26.1% | 73.9% | 100.0% |
| | Succeeded | Count | 285 | 498 | 783 |
| | | % within success | 36.4% | 63.6% | 100.0% |
| Total | | Count | 362 | 716 | 1078 |
| | | % within success | 33.6% | 66.4% | 100.0% |

Based on the results of all subjects, there were 285 successes and 77 failures in the case of achromatic images. Therefore, the odds of success are 285/77 = 3.701. Based on the results of all subjects, there were 498 successes and 218 failures in the case of colored images. Therefore, the odds of success are 498/218 = 2.284. By dividing the odds in the

case of colored and achromatic images, the odds ratio is 2.284/3.701 = 0.617. It means, that the odds of a subject to find the orientation of a colored figure is 0.617 times of the odds of finding the orientation of an achromatic image. In turn, the odds of succeeding in the achromatic test is 1/0.617 = 1.621 times of the chance of that in the colored test. Thus, the odds of success in achromatic images is higher than that in colored images.

The coefficient of subject × success × image is 0.218. Because of the loglinear model, this value was multiplied by eight, and we took the exponential of this value with base "e" according to Formula (4) (odds ratio = $e^{8 \times 0.218}$ = 5.72). The analysis of the cross-tabulation of the subject × success × image parameter was prepared for the validation of that (see Table 7).

**Table 7.** Cross-tabulation of subject × success × image parameters.

| Image | | | | Success | | Total |
|---|---|---|---|---|---|---|
| | | | | Failed | Succeeded | |
| Achromatic | Subject | Normal trichromat | Count | 72 | 252 | 324 |
| | | | % within subject | 22.2% | 77.8% | 100.0% |
| | | Anomalous trichromat | Count | 5 | 33 | 38 |
| | | | % within subject | 13.2% | 86.8% | 100.0% |
| | Total | | Count | 77 | 285 | 362 |
| | | | % within subject | 21.3% | 78.7% | 100.0% |
| Colored | Subject | normal trichromat | Count | 173 | 460 | 633 |
| | | | % within subject | 27.3% | 72.7% | 100.0% |
| | | anomalous trichromat | Count | 45 | 38 | 83 |
| | | | % within subject | 54.2% | 45.8% | 100.0% |
| | Total | | Count | 218 | 498 | 716 |
| | | | % within subject | 30.4% | 69.6% | 100.0% |

In the case of the achromatic series of images, the odds of success of anomalous trichromats are 33/5 = 6.6 and 252/72 = 3.5 for normal trichromats. In regard to the proportion of these two odds, the odds ratio is 6.6/3.5 = 1.88. This shows that, in the case of the achromatic series, anomalous trichromats find the orientation of the Landolt-C figures easier than normal trichromats. In the case of the colored series of images, the odds of success of anomalous trichromats are 38/45 = 0.844 and 460/173 = 2.66 for normal trichromats. Therefore, the odds ratio is 0.844/2.66 = 0.317. This shows that, in the case of the colored series, anomalous trichromats find the orientation of the Landolt-C figures with more difficulty. Comparing the odds ratios (1.88/0.317 = 5.93), it can be concluded that, in the case of achromatic images, the chance that anomalous trichromats find the orientation of the Landolt-C figure is 5.93-times higher than that for normal trichromats.

The estimation of the loglinear model is similar, as the value calculated of the model is 5.72, while the value of the chance ratio for success is 5.93 (5.93−5.72 = 0.21).

The estimate for the success × ordinal number parameter is −0.0353, hence even this parameter is significant. Its exhibitor based on *e* is 0.965. It can be stated that if the ordinal number increases with one (hence, the next, more difficult task appears) the chance of that the subject will succeed is 0.965-fold of the chance of failure. With other words as

the Landolt-C figures get more difficult to distinguish from the background the chance of success decreases by 3.5% in average, so the chance of failure increases.

### 3.2. Analysis of the Parameters of the Eye-Tracker

The parameters of the eye-tracker were ranked, and the average ranks were calculated in the case of ties. A generalized linear model was run, based on the new, ranked eye-tracker parameters. The SRH test was calculated on the results of the generalized linear model. In this test the *F*-test of the between subject effects is transformed to $\chi^2$-test and the *p*-value is corrected based on that. The SRH test is the non-parametric equivalent of the two-way ANOVA test, including the interaction terms as well.

Hereinafter the results are introduced in two parts regarding the applied two AOIs. The first part includes the eye-movement variables of the ring of the Landolt-C figure, as the follows: TTFF, time to first fixation; FFD, first fixation duration; FD, fixation duration; FC, fixation count; VD, visit duration; VC, visit count; and TTFMC, time to first mouse click. The second part details the following eye-movement variables of the hiatus of the Landolt-C figure: ttff2, FFD2, FD2, FC2, VD2 and VC2.

### 3.2.1. Analysis of the Eye Movements Corresponding to the Ring of the Landolt-C Figure

Only those parameters are displayed that significantly affect any of the parameters based on the SRH test (see Table 8).

**Table 8.** The significant results of the Scheirer–Ray–Hare (SRH) test on the parameters of eye movements corresponding to the ring of the Landolt-C figure.

| Parameter | Variable | H | *p* |
|---|---|---|---|
| Subject | Rank of FD | 6.197 | 0.013 |
| | Rank of FC | 10.816 | 0.001 |
| | Rank of VD | 7.293 | 0.007 |
| | Rank of VC | 3.873 | 0.049 |
| | Rank of TTFMC | 20.738 | <0.001 |
| Subject × success | Rank of TTFF | 7.462 | 0.006 |
| Subject × image | Rank of TTFF | 4.022 | 0.045 |
| | Rank of TTFMC | 8.555 | 0.003 |
| Success × image | Rank of TTFF | 4.080 | 0.043 |
| | Rank of FD | 8.649 | 0.003 |
| | Rank of FC | 7.171 | 0.007 |
| | Rank of VD | 6.872 | 0.009 |
| | Rank of VC | 10.036 | 0.002 |
| Subject × success × image | Rank of TTFF | 6.407 | 0.011 |
| | Rank of FD | 10.887 | <0.001 |
| | Rank of FC | 10.817 | 0.001 |
| | Rank of VD | 9.594 | 0.002 |
| | Rank of VC | 9.725 | 0.002 |
| | Rank of TTFMC | 8.522 | 0.004 |

TTFF, time to first fixation; FFD, first fixation duration; FD, fixation duration; FC, fixation count; VD, visit duration; VC, visit count; TTFMC, time to first mouse click.

Those combinations of variables were analyzed in detail that were significant even in the loglinear model (success × image and success × image × subject). Regarding the success × image × subject parameter, each variable is significant, excluding FFD.

Regarding the success × image parameter, each variable is significant, excluding FFD and TTFMC. The following cross-tabulation tables of image x success parameter considers the results of each subject. Time to first fixation was longer in the case of failure than in the case of success. The longer fixation time increased the chance of failure (see Table 9).

**Table 9.** Cross-tabulation of image x success for time to first fixation (TTFF).

| | | Image | |
|---|---|---|---|
| | | **Achromatic** | **Colored** |
| | | **TTFF Mean (s)** | **TTFF Mean (s)** |
| Success | Failed | 1.46 [a] | 1.21 [b] |
| | Succeeded | 0.96 [c] | 0.85 [d] |

The results of the Mann–Whitney statistical test are indicated by the letters in the superscript. If the letters (a,b,c,d) in the superscripts of two parameters are different, the two parameters are significantly ($\alpha = 0.05$) different from each other.

Fixation duration was always longer in the case of the colored image series than in the case of achromatic images. In the case of success, the duration of fixation decreased to the third part of that in the case of failure. The increase of the duration of fixation went hand in hand with the increase of the chance of failure (see Table 10).

**Table 10.** Cross-tabulation of image × success for fixation duration (FD).

| | | Image | |
|---|---|---|---|
| | | **Achromatic** | **Colored** |
| | | **FD Mean (s)** | **FD Mean (s)** |
| Success | Failed | 1.87 [a] | 2.95 [b] |
| | Succeeded | 0.58 [c] | 0.88 [d] |

The results of the Mann–Whitney statistical test are indicated by the letters in the superscript. If the letters (a,b,c,d) in the superscripts of two parameters are different, the two parameters are significantly ($\alpha = 0.05$) different from each other.

The visit duration was longer in the case of colored images, especially in the case of failure (see Table 11).

**Table 11.** Cross-tabulation of image x success for visit duration (VD).

| | | Image | |
|---|---|---|---|
| | | **Achromatic** | **Colored** |
| | | **VD Mean (s)** | **VD Mean (s)** |
| Success | Failed | 2.05 [a] | 3.35 [b] |
| | Succeeded | 0.66 [c] | 0.99 [d] |

The results of the Mann–Whitney statistical test are indicated by the letters in the superscript. If the letters (a,b,c,d) in the superscripts of two parameters are different, the two parameters are significantly ($\alpha = 0.05$) different from each other.

The following cross-tabulation tables of subject × image × success parameter consider even the two groups of subjects. Based on the analysis of the experienced correspondences, it can be stated that, in the case of failure, anomalous trichromats fixated the achromatic figures significantly longer. Normal trichromats fixated the colored images shorter in time when they succeeded than in the case of failure. In the case of achromatic images, there was no significant difference between subjects when they succeeded (see Table 12).

**Table 12.** Cross-tabulation of subject $\times$ success $\times$ image for fixation duration (FD).

| | | Image | | | |
|---|---|---|---|---|---|
| | | Achromatic | | Colored | |
| | | Failed | Succeeded | Failed | Succeeded |
| | | FD Mean (s) | FD Mean (s) | FD Mean (s) | FD Mean (s) |
| Subject | Normal trichromat | 1.80 [a,b] | 0.58 [c] | 3.20 [d] | 0.84 [c] |
| | Anomalous trichromat | 2.86 [a,d] | 0.60 [c] | 1.99 [a,b,d] | 1.38 [b] |

The results of the Mann–Whitney statistical test are indicated by the letters in the superscript. If the (a,b,c,d) letters in the superscripts of two parameters are different, the two parameters are significantly ($\alpha = 0.05$) different from each other. Each block (achromatic and colored) should be interpreted separately.

When failed, the visit duration of anomalous trichromats was longer in the case of achromatic images and shorter in the case of colored images than normal trichromats. Normal trichromats visited the colored images to a shorter time when they succeeded. In the case of achromatic images, there was no significant difference between subjects when they succeeded (see Table 13).

**Table 13.** Cross-tabulation of subject $\times$ success $\times$ image for visit duration (VD).

| | | Image | | | |
|---|---|---|---|---|---|
| | | Achromatic | | Colored | |
| | | Failed | Succeeded | Failed | Succeeded |
| | | VD Mean (s) | VD Mean (s) | VD Mean (s) | VD Mean (s) |
| Subject | Normal trichromat | 1.96 [a,b] | 0.66 [c] | 3.62 [d] | 0.95 [c] |
| | Anomalous trichromat | 3.24 [a,d] | 0.71 [c] | 2.30 [a,b,d] | 1.57 [b] |

The results of the Mann–Whitney statistical test are indicated by the letters in the superscript. If the letters (a,b,c,d) in the superscripts of two parameters are different, the two parameters are significantly ($\alpha = 0.05$) different from each other. Each block (achromatic and colored) should be interpreted separately.

In the case of the achromatic image series, when anomalous trichromats succeeded, their time to first fixation was longer than in the case of failure. For normal trichromats, this was the other way around. The time to first fixation of colored images was longer in the case of false answers than in the case of right answers; it was shorter both for normal and anomalous trichromats. In the case of achromatic images and failure, the time to first fixation of normal trichromats was three times longer than that of anomalous trichromats (see Table 14).

3.2.2. Analysis of the Eye Movements Corresponding to the Hiatus of the Landolt-C Figure

In the case of the parameters regarding to the hiatus of the Landolt-C figure, less cases of significant effects were found. Three variables affected the subject parameter significantly (TTFF2, FFD2 and FC2), two variables affected the subject $\times$ success parameter significantly (FD2 and VD2) and only one variable (TTFF2) affected both subject $\times$ image and success $\times$ image $\times$ subject parameters (see Table 15).

**Table 14.** Cross-tabulation of subject × success × image for time to first fixation (TTFF).

| | | Image | | | |
|---|---|---|---|---|---|
| | | Achromatic | | Colored | |
| | | Failed | Succeeded | Failed | Succeeded |
| | | TTFF Mean (s) | TTFF Mean (s) | TTFF Mean (s) | TTFF Mean (s) |
| Subject | Normal trichromat | 1.52 [a] | 0.94 [b,c] | 1.19 [d,e] | 0.84 [f] |
| | Anomalous trichromat | 0.55 [b,d,f] | 1.11 [a,b,d] | 1.26 [a,c,e] | 0.92 [b,d,f] |

The results of the Mann–Whitney statistical test are indicated by the letters in the superscript. If the letters (a,b,c,d,e,f) in the superscripts of two parameters are different, the two parameters are significantly ($\alpha = 0.05$) different from each other. Each block (achromatic and colored) should be interpreted separately.

**Table 15.** The significant results of the SRH test on the parameters of eye movements corresponding to the hiatus of the Landolt-C figure.

| Parameter | Variable | H | *p* |
|---|---|---|---|
| Subject | Rank of TTFF2 | 13.017 | <0.001 |
| | Rank of FFD2 | 5.013 | 0.025 |
| | Rank of FC2 | 4.760 | 0.029 |
| Subject × success | Rank of FD2 | 5.329 | 0.021 |
| | Rank of VD2 | 5.166 | 0.023 |
| Subject × image | Rank of TTFF2 | 7.873 | 0.005 |
| Subject × success × image | Rank of TTFF2 | 6.949 | 0.008 |

When anomalous trichromats succeeded, their time to first fixation was more than twice as long for achromatic images; however, for colored images, the time before the first fixation was much shorter than that of normal trichromats. It took less time for normal trichromats than for anomalous trichromats to fixate on the hiatus in the case of achromatic images and success. In the case of colored images, anomalous trichromats needed shorter time to first fixation when they succeeded than when they failed (see Table 16).

**Table 16.** Cross-tabulation of subject × success × image for time to first fixation (TTFF2).

| | | Image | | | |
|---|---|---|---|---|---|
| | | Achromatic | | Colored | |
| | | Failed | Succeeded | Failed | Succeeded |
| | | TTFF2 Mean (s) | TTFF2 Mean (s) | TTFF2 Mean (s) | TTFF2 Mean (s) |
| Subject | Normal trichromat | 1.52 [a,b] | 0.91 [d] | 1.05 [b] | 1.86 [c] |
| | Anomalous trichromat | 1.58 [a,b,c] | 1.90 [c] | 1.60 [a,c] | 1.26 [a] |

The results of the Mann–Whitney statistical test are indicated by the letters in the superscript. If the letters (a,b,c,d) in the superscripts of two parameters are different, the two parameters are significantly ($\alpha = 0.05$) different from each other. Each block (achromatic and colored) should be interpreted separately.

### 3.2.3. The Survival Analysis of the Success of Subjects

With the survival analysis, the chance of success decrease is observed in the function of the ordinal number of the images shown to the subjects. Anomalous and normal trichromats were considered separately (see Figures 2 and 3). In this analysis, the survival

function shows the probability of success in the function of the ordinal number of the images. The observed event happened when the subject failed to find the hiatus of the Landolt-C figure. The first value of the survival function belongs to 0 that refers to the beginning of the observation. At this point obviously none of the subjects failed (or succeeded) to find the hiatus of the Landolt-C figure; therefore, the function takes its maximal value: 1 (in terms of probability of success: 100%).

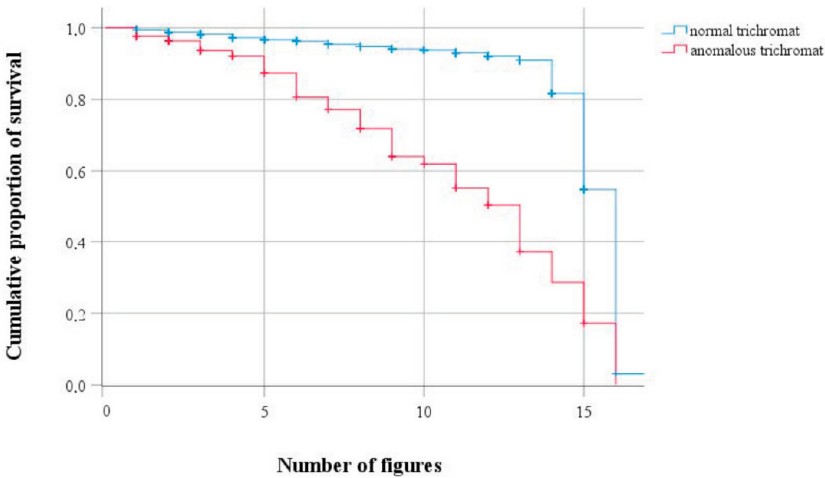

**Figure 2.** Survival functions of anomalous trichromats and normal trichromats for the colored image series.

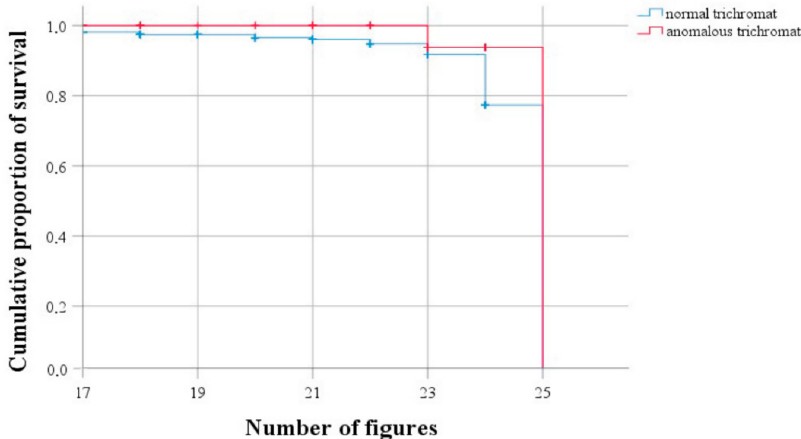

**Figure 3.** Survival functions of anomalous trichromats and normal trichromats for the achromatic image series.

Whenever an event happens, i.e., the subject fails to detect the hiatus, the value of the survival function decreases. The "+" sign denotes that the observed event did not happen (censored), the subject succeeded to find the hiatus and the function keeps its current value. Figures 2 and 3 show the survival functions calculated for the achromatic and colored image series, respectively.

As it is shown in Figure 2 and Table 17, in the case of the colored image series, anomalous trichromats (red plot) were significantly less successful in finding the hiatus of the Landolt-C than normal trichromats (blue plot). The decrease of the survival function of the anomalous trichromats is constantly steeper than that of the normal trichromats regarding the first 13 images. After the ordinal number 13, a steeper decrease can be detected even in the survival function of the normal trichromats. A final cutoff to zero was predestinated, since the last element of the image series was designed, as neither normal nor anomalous trichromats could find the orientation of the Landolt-C figure; the chromatic

difference between the background and the Landolt-C figure reached and even exceeded the limit of the just noticeable difference both for anomalous and normal trichromats.

**Table 17.** Comparison of the survival functions of achromatic- and colored-image series.

| | Achromatic Images | | |
|---|---|---|---|
| | $\chi^2$ | df | *p* |
| Log Rank (Mantel–Cox) | 1.531 | 1 | 0.216 |
| Breslow (Generalized Wilcoxon) | 1.514 | 1 | 0.218 |
| Tarone–Ware | 1.599 | 1 | 0.206 |
| | Colored Images | | |
| Log Rank (Mantel–Cox) | 84.706 | 1 | <0.001 |
| Breslow (Generalized Wilcoxon) | 83.086 | 1 | <0.001 |
| Tarone–Ware | 91.747 | 1 | <0.001 |

df, degree of freedom.

Figure 3 shows that the relations of the two survival functions changed in the case of achromatic image series. In this case, the survival function of anomalous trichromats (red plot) exceeds that of normal trichromats (blue plot). Even though Table 17 reports that the result of this comparison is not significant, the graph shows that the first failure of anomalous trichromats occurred only at the ordinal number 23 (the eighth image in the achromatic series), while the survival function of normal trichromats decreases from value to value; even in the last two steps, the decrease of the survival function is steeper for the normals than that for the anomalous trichromats. The final cutoff to 0 value was predestinated just as explained in considering the colored-image series. Therefore, if we consider the range in which the task was possible to execute successfully for subjects, this graph shows that, in the case of the observed achromatic images, anomalous trichromats were more successful than normal trichromats.

## 4. Discussion

As no works on the combination of eye-tracking method and pseudoisochromatic tests have been published so far, it is difficult to compare specific research results.

Based on our results an effective way of evaluating the multiple or cross-classified contingency tables is loglinear regression; however, this method shows only which parameter is significant and which is not [47]. In case the conditions of the parametric statistical tests are not appropriate (data type, homogeneity of variance and normal distribution), we can either perform transformation on our data or apply the corresponding non-parametric method. The Scheirer–Ray–Hare (SRH) test is a non-parametric pair of ANOVA; therefore, it is an appropriate statistical method to describe and compare the parameters of eye-tracking [48,49].

Survival analysis, used primarily in clinical research, has been used successfully in several cases, in combination with eye-movement tracking in relation to consumer choices [50]. It can be well observed that each discipline adapts different methodologies.

The aim of our first test series, the colored test series, was to compare our results with the classical color-vision testing methods. The survival analysis of the results of the colored test series showed that normal trichromats performed more successfully, which corresponds well with the literature regarding loss of chromatic discrimination ability of anomalous trichromats [51].

For further analysis, we applied the second test series, achromatic test series, measuring luminance contrast sensitivity. Even though our analysis did not show significant evidence on superior achromatic discrimination of anomalous trichromats compared to normal trichromats, the results of survival analysis suggested that they performed better. Similar results can be found in the current literature [52].

## 5. Conclusions

The main question of our research was whether anomalous and normal trichromats can be distinguished based on eye movements under different conditions of luminance and chromatic contrast sensitivity. Following our research questions, we concluded the followings.

1.  Which statistical method is suitable to model the differences between the number of right answers of anomalous and normal trichromats? The differences between the answers of anomalous trichromats and normal trichromats can be modeled with statistical methods. The right answers of anomalous trichromats and normal trichromats can be described with multiple or cross-classified contingency tables. For comparing these parameters, Scheirer–Ray–Hare (SRH) test can be applied.

2.  Which of eye movement variables have significant effect on the parameters (image, success and subject) and their interactions? We identified the variables of the eye tracking that have significant effect on the parameters (image, success, subject and ordinal number). In the case of the observation of Landolt-C figures, the variables related to fixation duration, fixation count, visit duration or visit count significantly affected the subject, success or image parameters and their interactions. In the case of observing the hiatus of the Landolt-C figures, the time to first fixation was significant, meaning that, if someone recognizes the hiatus, he or she looks at that instantly.

3.  Is there any significant difference between eye movement variables of normal and anomalous trichromats? The estimate for the success x ordinal number parameters validated the design of the test: As the Landolt-C figures get more difficult to distinguish from the background, the chance of success decreases by 3.5% in average, so the chance of failure increases.

4.  How does the survival time of anomalous and normal trichromats change in the case of achromatic and colored figures? Besides the results of the eye tracker variables, differences between normal and anomalous trichromats was traced also. In the case of achromatic image series, the chance of success for anomalous trichromats is 5.93-fold than that for normal trichromats. The survival functions of normal and anomalous trichromats go in different ways in the case of colored and achromatic images. Based on the results of the survival analysis, normal trichromats are more successful in colored images; that was proven with statistical methods as well. Conversely, the survival function of anomalous trichromats shows that they are more successful in achromatic images that can be explained with the assumption that their luminance contrast sensitivity is better, as they have to be able to discriminate smaller luminance differences in everyday life.

The limitation of our study was the low number of anomalous trichromats, even though the applied loglinear analysis handles the asymmetry between the two groups well (normal and anomalous trichromats).

Note the importance of the experimental design, including both the environmental parameters (such as lighting conditions and the relative position of the stimuli and the subject) and the visual stimuli (such as the appropriate colorimetric design of the figures and calibration of the display following corresponding standards).

Based on our conclusions, we recommend applying eye-tracking methods in combination with pseudoisochromatic tests, for testing color-vision deficiency, in practice.

**Author Contributions:** All authors have contributed to the research. Conceptualization, L.S., K.S., K.W.; investigation, L.S. and Á.N.; data curation A.G. and S.K.; methodology, A.G. and S.K., software, S.K., A.G.; writing—original draft, S.K., L.S. and A.G.; writing—review and editing, Á.N. and Á.U.; visualization S.K.; supervision, Z.K., K.S. and K.W.; project administration, Á.N. and Á.U.; funding acquisition, L.S., S.K. and K.S. All authors have read and agreed to the published version of the manuscript.

**Funding:** This project was supported by the János Bolyai Research Scholarship of the Hungarian Academy of Sciences. This work was partly supported by the European Union and co-financed by

the European Social Fund (grant agreement no. EFOP-3.6.3-VEKOP-16-2017-00005). The research reported in this paper was supported by the "TKP2020, Institutional Excellence Program" of the National Research Development and Innovation Office in the field of Biotechnology research area. Attila Gere thanks the support of the Premium Postdoctoral Research Program and the support of National Research, Development and Innovation Office of Hungary (OTKA, contracts No. K134260). The research reported in this paper was supported by the NRDI Fund (TKP2020 NC, Grant No. BME-NCS) based on the charter of bolster issued by the NRDI Office under the auspices of the Ministry for Innovation and Technology. The research reported in this paper was supported by the NRDI Fund (TKP2020 IES, Grant No. BME-IE-BIO), based on the charter of bolster issued by the NRDI Office under the auspices of the Ministry for Innovation and Technology.

**Institutional Review Board Statement:** The study was conducted according to the guidelines of the Declaration of Helsinki, and approved by the Institutional Review Board (or Ethics Committee) of United Ethical Review Committee for Research in Psychology (protocol code 16/2016, date of approval: 10.02.2016.).

**Informed Consent Statement:** Informed consent was obtained from all subjects involved in the study.

**Data Availability Statement:** Not applicable.

**Acknowledgments:** Á.N. expresses his gratitude to the Doctoral School of Food Science of Hungarian University of Agriculture and Life Sciences.

**Conflicts of Interest:** The authors declare no conflict of interest.

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
