# Peer review of "Eye-Tracker Analysis of the Contrast Sensitivity of Anomalous and Normal Trichromats: A Loglinear Examination with Landolt-C Figures"

_applsci, doi:10.3390/app11073200_

Round 1

Reviewer 1 Report

This study explores whether anomalous and normal trichromats could be distinguished based on eye movements. The study may have practical application in the food area, industry and trade, and other fields. The text needs a general cleanup to remove typos and some incongruencies. More information is needed in some parts of the text.

Introduction

Line 52-53: "Not only contrast helps our eye in orientation. The image processing functions of our brain play an important role as well [4-6]" these two statements are not clear, rephrase.

Line 54: All the references shown have more than 4 years; delete "current".

Line 56: References [15-17] should be at the end of the statement.

Line 62: Considering that precise and accurate are not synonymous, maybe the authors want to say accurate.

Line 64: Reference [20-23] have more than 15 years; authors should make an effort to use more recent references (or a review paper).

Line 67-69: Finish the statement with examples of "the wide range of measured parameters".

Line 69-72: This statement is too long (4 lines without any punctuation) and has redundant information; it should be rephrased.

Line 73-74: The eye-tracker records the eye movements, doesn't measure luminance or colour. Thus, the statement should be rephrased according to this (e.g., our research's main question was whether anomalous and normal trichromats could be distinguished based on eye movements under different conditions of luminance and chromatic contrast sensitivity.)

Same comment on the first statement of the Conclusion section (line 419-421).

Methods

Line 110: references [27, 28] should be in the main text

Line 127: unknown symbol between sRGB and XYZ

Line 141: Add the mean age ± standard deviation of subjects.

Line 141: replace "9" with "Nine".

Line 141-143: Is seems that anomaloscopic measures were just done for the 9 subjects reporting colour visions deficiency. If this is true, how can authors guarantee that the other 81 subjects have normal colour vision? This point must be clarified

Besides, colour vision deficiency is a very generic term; give more detail about the nine subjects' colour vision deficiencies.

It seems that subjects' visual health was not tested and this should be stated and explored in the study's limitation. For example, ametropias may affect contrast sensitivity.

There is also a significant asymmetry between normal (81) and abnormal (9) colour vision subjects. How well the statistical models used by the authors lead with this?

Line 188: Replace "The estimated model…" with "The more parsimonious model obtained …".

Line 242-244: fixe the brackets  

Results

302-305: I recommend making the abbreviations in capital letter, and instead of being like

(line 302) "the followings: ttff: time to first fixation, ffd: first fixation duration …" do like "the followings: time to first fixation (TTFF); first fixation duration (FFD); …" and repair all text and tables according to this.  

Line 317: It is easier to follow if "elapsed time until the first fixation" were replaced by "time to first fixation" like it was previously defined (line 302).

Line 320: In this Table and the following Tables, all numbers have a superscript; what is the utility of these superscripts in the context of this draft?

Line 321: It is easier to follow if "Fixation" be replaced by "Fixation duration" like it was previously defined (line 302).

Line 330: rectify “(vd)).”

Line 334: considerate to remove the bold words from all text. If you want to call attention, use the abbreviations defined in line 302-305

Line 335: refrain from using significantly/significant in the text (all parts) when comparisons are not supported by statistics (post-hoc), i.e, this word should be reserved for p-values to avoid ambiguously.

Line 302: "time to the first fixation" and lines 350 and 370: "time for first fixation": uniformize

Conclusions

Limitations of this study should be addressed.

Line 419-420:  see the previous comment about this statement

Line 425: rectify "resression"

Line 431: rectify "succes"

Author Response

Answers for review of

Manuscript ID: applsci-1151309

submitted to Applied Sciences

entitled

”Eye-Tracker Analysis of the Contrast Sensitivity of Anomalous and Normal Trichromats: a Loglinear Examination with Lan-dolt-C Figures”

Authors would like to thank for the consideration and evaluation of the submitted manuscript and for making time to read and review it. The corrections and comments were very useful and helped a lot to improve the manuscript. We have carefully read through the recommendations and have made the appropriate corrections. The whole manuscript has also been proof-read for all English discrepancies. Below is point by point response to the reviewers.

Reviewer #1

This study explores whether anomalous and normal trichromats could be distinguished based on eye movements. The study may have practical application in the food area, industry and trade, and other fields. The text needs a general cleanup to remove typos and some incongruencies. More information is needed in some parts of the text.

Introduction

  1. Line 52-53: "Not only contrast helps our eye in orientation. The image processing functions of our brain play an important role as well [4-6]" these two statements are not clear, rephrase.
    We have rephrased this sentence: ”Vision is created by the contrast sensitivity of the eye and the image processing of the brain.”

  2. Line 54: All the references shown have more than 4 years; delete "current".
    We have changed „current” to „important”, and included a corresponding paper from 2021.

  3. Line 56: References [15-17] should be at the end of the statement.
    We have replaced the references.

  4. Line 62: Considering that precise and accurate are not synonymous, maybe the authors want to say accurate.
    Thank you for the suggestion. We have clarified this and changed to accurate.

  5. Line 64: Reference [20-23] have more than 15 years; authors should make an effort to use more recent references (or a review paper).
    Thank you for the comment, we have supplemented the references with the following review papers:

Hasrod N, Rubin A. Colour vision: A review of the Cambridge Colour Test and other colour testing methods. Afr Vision Eye Health 2015, 74 (1), 1–7. http://dx.doi.org/10.4102/ aveh.v74i1.23

Zarazaga, A. F., Vásquez, J. G., & Royo, V. P. (2019). Review of the main colour vision clinical assessment tests. Archivos de la Sociedad Española de Oftalmología (English Edition)94(1), 25-32. https://doi.org/10.1016/j.oftale.2018.08.010”

  1. Line 67-69: Finish the statement with examples of "the wide range of measured parameters".
    We have clarified with the following examples: „Observation of observers’ gazing behaviour with eye tracking is a widely used method that can provide detailed analysis based on the wide range of measured parameters, such as the count or the duration of fixations or the decision time.”

  1. Line 69-72: This statement is too long (4 lines without any punctuation) and has redundant information; it should be rephrased.

We have shortened the long sentence in order to avoid redundant information: „We haves shortened and clarified this sentence: Therefore, the aim of this research was to answer whether eye tracking can be a method for testing color vision deficiency.”

  1. Line 73-74: The eye-tracker records the eye movements, doesn't measure luminance or colour. Thus, the statement should be rephrased according to this (e.g., our research's main question was whether anomalous and normal trichromats could be distinguished based on eye movements under different conditions of luminance and chromatic contrast sensitivity.)
    Thank you for the suggestion, we have implemented.

  1. Same comment on the first statement of the Conclusion section (line 419-421).
    Thank you for the suggestion, we have implemented.

 Methods

  1. Line 110: references [27, 28] should be in the main text
    We have replaced references to the first sentence of ’1.1. Visual stimuli’.

  2. Line 127: unknown symbol between sRGB and XYZ
    We have corrected to: „sRGB to XYZ conversion”

  3. Line 141: Add the mean age ± standard deviation of subjects.
    Thank you for the recommendation. We have rephrased and completed the sentence: „90 subjects (42 male, 48 female) aged between 18 and 35 (25.5 ± 3.6 years) participated in the study.

  1. Line 141: replace "9" with "Nine".
    We have replaced.

  2. Line 141-143: Is seems that anomaloscopic measures were just done for the 9 subjects reporting colour visions deficiency. If this is true, how can authors guarantee that the other 81 subjects have normal colour vision? This point must be clarified
    We have completed the paragraph: ”Normal and defective colour vision was confirmed with anomaloscopic measurements: R/G scores within the range of 40±5 indicated normal colour vision while R/G scores ex-cluding this range indicated defective colour vision. Eighty-one of the subjects were nor-mal trichromats and nine of the subjects reported colour vision deficiency which was con-firmed with the anomaloscopic measurements.”

  3. Besides, colour vision deficiency is a very generic term; give more detail about the nine subjects' colour vision deficiencies.
    We have inserted: ”Since in many cases subjects with deuteranomaly have similar results on the applied pseudoisochromatic tests than of normal trichromats, we excluded deuteranomalous subjects and we invited only subjects with protanomaly”.

  4. It seems that subjects' visual health was not tested and this should be stated and explored in the study's limitation. For example, ametropias may affect contrast sensitivity.
  5. We have clarified: "Visual acuity of the subjects was monitored in a report and tested with the first test figures of both test series. Since the field of view under which the detectable stimulus was shown did not change during the test, we used the first images of the two test series. In case subjects could clearly identify the orientation of both Landolt-C characters, we accepted that their visual acuity is sufficient for participation."

  6. There is also a significant asymmetry between normal (81) and abnormal (9) colour vision subjects. How well the statistical models used by the authors lead with this?

Thank you for your recommendation, we have added explanation: “Moving from symmetrical to asymmetrical, we applied loglinear analysis. This analysis models the log odds that a given observation (fail or success) will appear in one category (normal trichromats) of a dependent variable relative to other categories (anomalous trichromats). The purpose of the analysis is to model the asymmetry exist in the data and translate it in term of odds.”

  1. Line 188: Replace "The estimated model…" with "The more parsimonious model obtained …".
    We have replaced.

  2. Line 242-244: fixe the brackets  
    We have fixed.

 Results

  1. 302-305: I recommend making the abbreviations in capital letter, and instead of being like

(line 302) "the followings: ttff: time to first fixation, ffd: first fixation duration …" do like "the followings: time to first fixation (TTFF); first fixation duration (FFD); …" and repair all text and tables according to this.  
Thank you for your recommendation, we have corrected it.

  1. Line 317: It is easier to follow if "elapsed time until the first fixation" were replaced by "time to first fixation" like it was previously defined (line 302).
    Thank you for your recommendation, we have changed it.

  2. Line 320: In this Table and the following Tables, all numbers have a superscript; what is the utility of these superscripts in the context of this draft?
    We have added the following explanation to each reliable table: ”The results of the statistical test are indicated by the letters in the superscript. If the letters in the superscripts of two parameters are different, the two parameters are significantly (α=0.05) different from each other.”

  3. Line 321: It is easier to follow if "Fixation" be replaced by "Fixation duration" like it was previously defined (line 302).

We have replaced.

  1. Line 330: rectify “(vd)).”
    We have corrected.

  2. Line 334: considerate to remove the bold words from all text. If you want to call attention, use the abbreviations defined in line 302-305
    Bold words have been removed from the text everywhere.

  3. Line 335: refrain from using significantly/significant in the text (all parts) when comparisons are not supported by statistics (post-hoc), i.e, this word should be reserved for p-values to avoid ambiguously.
    We have added to the relevant tables the p-values to the Z-scores.

  4. Line 302: "time to the first fixation" and lines 350 and 370: "time for first fixation": uniformize
    We have changed „time for first fixation” to „time to first fixation” everywhere.

Conclusions

  1. Limitations of this study should be addressed.
    The conclusions have been supplemented with limitations.

  2. Line 419-420:  see the previous comment about this statement
    Thank you for the suggestion, we have implemented.

  3. Line 425: rectify "resression"
    We have corrected.

  4. Line 431: rectify "succes"
    We have corrected.

Reviewer 2 Report

Summary

This manuscript has great potential, well organized and uses useful tables to quickly summarize the results. Although this document should address a number of important issues to improve mainly because of its limitation in structure as a scientific article.

Specific Comments

  1. The introduction: Authors need to work to improve the development and explanation of introductory concepts, such as visual acuity, objects, contrast, processing functions, function or different color vision alterations.

  • The objective or hypothesis of this study is not introduced and what value it would have in daily clinical practice.

  1. The methodology is one of the most important aspects in research and this manuscript has multiple deficiencies, such as lack of greater description and details of methodological design and improving specifications in the sample description:

  • Line 141. No standard deviation (SD) of sample age is observed.
  • Criteria that have been followed to include or exclude subjects from the sample and were not susceptible in both cases are not included.
  • There is no description or visual measurement criteria for the assessment of a possible refractive error that can influence the results.
  • The authors do not specifically describe what kind of actual lighting conditions they used to perform the measurements, only that it was controlled.
  • The manuscript does not include which eye or visual conditions they used to perform the different tests (monocular, biocular, binocular) and was chosen randomly or under some conditions.
  • The authors did not provide the duration of the sessions and rest times.
  • The authors did not perform a specific discussion section in this manuscript, being one of the most important parts of a scientific essay. Relationships and generalizations from the results should be presented, the absence of connections should be mentioned and some points clarified and results interpreted with other previously published studies.
  • The study does not clearly conclude with the alleged starting hypothesis.

3st General Comments

The authors did a good job, but the manuscript needs to improve in aspects such as incomplete introduction, lack of data from the methodological section and mainly the discussion section that interprets and extrapolates all the results and is non-existent.

Author Response

Answers for review of

Manuscript ID: applsci-1151309

submitted to Applied Sciences

entitled

”Eye-Tracker Analysis of the Contrast Sensitivity of Anomalous and Normal Trichromats: a Loglinear Examination with Lan-dolt-C Figures”

Authors would like to thank for the consideration and evaluation of the submitted manuscript and for making time to read and review it. The corrections and comments were very useful and helped a lot to improve the manuscript. We have carefully read through the recommendations and have made the appropriate corrections. The whole manuscript has also been proof-read for all English discrepancies. Below is point by point response to the reviewers.

Reviewer #2

Summary

This manuscript has great potential, well organized and uses useful tables to quickly summarize the results. Although this document should address a number of important issues to improve mainly because of its limitation in structure as a scientific article.

Specific Comments

  1. The introduction: Authors need to work to improve the development and explanation of introductory concepts, such as visual acuity, objects, contrast, processing functions, function or different color vision alterations.

The Introduction section has been supplemented accordingly.

  1. The objective or hypothesis of this study is not introduced and what value it would have in daily clinical practice.

To date, a number of pseudo-isochromatic tests have been developed to diagnose colour vision deficiency, and anomaloscope instrument is used in clinical practice. The focus of our research was on whether the method of eye movement tracking could be suitable for the study of colour vision deficiency. Furthermore, the location and role of color vision studies are presented in the introduction section. The introduction has been supplemented with: „Therefore, the aim of this research was to answer whether eye tracking can be a method for testing color vision deficiency?”

  1. The methodology is one of the most important aspects in research and this manuscript has multiple deficiencies, such as lack of greater description and details of methodological design and improving specifications in the sample description:

The material and method section has been expanded and clarified in detail, with special regard to the reviewers' comments and suggestions.

  1. Line 141. No standard deviation (SD) of sample age is observed.

We have inserted details to 2.1.2. Subjects: „90 subjects (42 male, 48 female) aged between 18 and 35 (25.5 ± 3.6 years) participated in the study.”

  1. Criteria that have been followed to include or exclude subjects from the sample and were not susceptible in both cases are not included.

We have completed the paragraph 2.1.2. Subjects: ”Normal and defective colour vision was confirmed with anomaloscopic measurements: R/G scores within the range of 40±5 indicated normal colour vision while R/G scores ex-cluding this range indicated defective colour vision. Eighty-one of the subjects were nor-mal trichromats and nine of the subjects reported colour vision deficiency which was con-firmed with the anomaloscopic measurements.”

  1. There is no description or visual measurement criteria for the assessment of a possible refractive error that can influence the results.

We have clarified in 2.1.2. Subjects: "Visual acuity of the subjects was monitored in a report and tested with the first test figures of both test series. Since the field of view under which the detectable stimulus was shown did not change during the test, we used the first images of the two test series. In case subjects could clearly identify the orientation of both Landolt-C characters, we accepted that their visual acuity is sufficient for participation."

  1. The authors do not specifically describe what kind of actual lighting conditions they used to perform the measurements, only that it was controlled.
    We have inserted the following explanation: „Lighting conditions followed the recommendations of the ISO 3664 and ISO 12646 Standards. Approximately homogeneous luminance of 300lx was provided with a special attention on excluding glare and disturbing visual elements in the field of view.
  2. The manuscript does not include which eye or visual conditions they used to perform the different tests (monocular, biocular, binocular) and was chosen randomly or under some conditions.

We have included to 2.1.3. Eye-Tracking Procedure: „Measurements were executed binocularly.”

  1. The authors did not provide the duration of the sessions and rest times.

Two pictures were used as warm-up to familiarise the subjects with the procedure that were not included in the data analysis. This took an average of 4-6 seconds. The duration of responses to pseudo-isochromatic images was participant-dependent. On average 25-35 sec was spent in total. There was no break between the familiarision, the chromatic and the achromatic sessions. We have involved this explanation to the 2.1.1. Visial stimuli section.

  1. The authors did not perform a specific discussion section in this manuscript, being one of the most important parts of a scientific essay. Relationships and generalizations from the results should be presented, the absence of connections should be mentioned and some points clarified and results interpreted with other previously published studies.

We have included discussion highlighting relations with previously published studies.

  1. The study does not clearly conclude with the alleged starting hypothesis.

The conclusion has been supplemeneted and concisely restructured.

3st General Comments

The authors did a good job, but the manuscript needs to improve in aspects such as incomplete introduction, lack of data from the methodological section and mainly the discussion section that interprets and extrapolates all the results and is non-existent.

Authors would like to thank for the consideration and evaluation of the submitted manuscript and for making time to read and review it. Corrections, comments and suggestions were very useful and helped a lot to improve the manuscript.

Round 2

Reviewer 2 Report

Comments solved